# Methodological Review of Classification Trees for Risk Stratification: An Application Example in the Obesity Paradox

**DOI:** 10.3390/nu17111903

**Published:** 2025-05-31

**Authors:** Javier Trujillano, Luis Serviá, Mariona Badia, José C. E. Serrano, María Luisa Bordejé-Laguna, Carol Lorencio, Clara Vaquerizo, José Luis Flordelis-Lasierra, Itziar Martínez de Lagrán, Esther Portugal-Rodríguez, Juan Carlos López-Delgado

**Affiliations:** 1IRBLLeida (Institut de Recerca Biomèdica de Lleida Fundació Dr. Pifarré), Av. Alcalde Rovira Roure, 80, 25198 Lleida, Spain; lserviag@gmail.com (L.S.); mbadia26@gmail.com (M.B.); 2NUTREN-Nutrigenomics, Department of Experimental Medicine, University of Lleida, 25198 Lleida, Spain; josecarlos.serrano@udl.cat; 3Intensive Care Department, Hospital Universitario Germans Trias i Pujol, Carretera de Canyet, s/n, 08916 Badalona, Spain; luisabordeje@gmail.com; 4Intensive Care Department, Hospital Universitari Josep Trueta, Av. de França, s/n, 17007 Girona, Spain; carol_lorencio@hotmail.com; 5Intensive Care Department, Hospital Universitario de Fuenlabrada, Cam. del Molino, 2, 28942 Fuenlabrada, Spain; clara.vaquerizo@salud.madrid.org; 6Intensive Care Department, Hospital Universitario 12 de Octubre, Av. de Córdoba s/n, 28041 Madrid, Spain; makalyconru@hotmail.com; 74i+12 (Instituto de Investigación Sanitaria Hospital 12 de Octubre, Research Institute Hospital 12 de Octubre), Av. de Córdoba s/n, 28041 Madrid, Spain; 8Intensive Care Department, Hospital de Mataró, 08304 Mataró, Spain; itziarmz@hotmail.com; 9Intensive Care Department, Hospital Clínico Universitario de Valladolid, Av. Ramón y Cajal, 3, 47003 Valladolid, Spain; esther_burgos@hotmail.com; 10Area de Vigilancia Intensiva, Clinical Institute of Internal Medicine & Dermatology (ICMiD), Hospital Clínic de Barcelona, C/Villarroel, 170, 08036 Barcelona, Spain; juancarloslopezde@hotmail.com

**Keywords:** classification trees, machine learning, prediction modelling, intensive care unit, obesity paradox

## Abstract

Background: Classification trees (CTs) are widely used machine learning algorithms with growing applications in clinical research, especially for risk stratification. Their ability to generate interpretable decision rules makes them attractive to healthcare professionals. This review provides an accessible yet rigorous overview of CT methodology for clinicians, highlighting their utility through a case study addressing the “obesity paradox” in critically ill patients. Methods: We describe key methodological aspects of CTs, including model development, pruning, validation, and classification types (simple, ensemble, and hybrid). Using data from the ENPIC (Evaluation of Practical Nutrition Practices in the Critical Care Patient) study, which assessed artificial nutrition in ICU (intensive care unit) patients, we applied various CT approaches—CART (classification and regression trees), CHAID (chi-square automatic interaction detection), and XGBoost (extreme gradient boosting)—and compared them with logistic regression. SHAP (SHapley Additive exPlanation) values were used to interpret ensemble models. Results: CTs allowed for identification of optimal cut-off points in continuous variables and revealed complex, non-linear interactions among predictors. Although the obesity paradox was not confirmed in the full cohort, CTs uncovered a specific subgroup in which obesity was associated with reduced mortality. The ensemble model (XGBoost) achieved the best predictive performance (highest area under the ROC curve), though at the expense of interpretability. Conclusions: CTs are valuable tools in clinical epidemiology, complementing traditional models by uncovering hidden patterns and enhancing risk stratification. While ensemble models offer superior predictive accuracy, their complexity necessitates interpretability techniques such as SHAP. CT-based approaches can guide personalized medicine but require cautious interpretation and external validation.

## 1. Introduction

In medical decision-making, tools are needed to assist in risk stratification for the occurrence of adverse events such as death or complications, as well as for making accurate diagnoses [1]. The development of such tools requires the application of various methodologies for risk model construction. Numerous approaches can be employed, ranging from classical statistical methods to advanced machine learning techniques [1]. Among these, classification tree (CT)-based methodologies remain an appealing option due to their specific advantages [2].

This work aims to provide an overview of the methodology of classification trees (CT), offering a perspective directed at clinical professionals interested in risk models. A technical yet accessible language has been used so that the content can be understood without requiring extensive methodological knowledge. For readers interested in a deeper understanding of the topic, we provide bibliographic references that we consider most suitable for further exploration, including examples related to nutrition-related problems.

Our focus is limited to classification trees that generate classification rules essential for establishing relationships between variables and identifying groups of patients with specific characteristics. CTs belong to a family of machine learning algorithms that use tree-like structures to support decision-making. This review is accompanied by a real-data application example to illustrate the utility and features of CTs.

### 1.1. Concept of a Classification Tree

A CT is the graphical representation of a series of classification rules. Starting from a root node, which includes all cases, the tree branches into different “child” nodes containing subgroups of cases. The splitting criterion, also known as branching criterion, is optimally determined after examining the values of all included predictor variables [2]. CTs are a form of supervised machine learning in which the algorithm is provided with records that include predictor variables and the outcome variable. These algorithms function by reducing classification error until the optimal CT is found [3].

CT methodology has been in use for quite some time. The earliest references to CTs are attributed to Quinlan in 1986 [4]. Regression trees, which use continuous outcome variables, had already been in use for over 50 years by then [3].

### 1.2. Phases in the Construction of a Classification Tree

To illustrate the process of constructing a CT, we use the CART (classification and regression tree) model as a reference. This process can be divided into several phases [5]:Phase 1—Tree Development: From the root node, the most appropriate variable is identified to split the node into two child nodes by establishing an optimal cut-off point if the variable is continuous. Each child node is subsequently split following the same methodology. A supervised machine learning model is used, with all records including predictor variables and the outcome variable submitted to the algorithm.Phase 2—Tree Growth Stopping Criteria: Tree development can continue until terminal nodes contain only a single case, or when the value of the dependent variable is the same for all cases within a node. Additional criteria, such as a minimum number of cases per node, can be defined to prevent excessive branching.Phase 3—Tree Pruning: A CT developed using the aforementioned method tends to be overly complex and branched, which may lead to overfitting the training dataset. Removing superfluous branches results in a simpler tree with better generalizability. The pruning process uses predefined cost–complexity criteria to eliminate branches that add more complexity than effectiveness. Supervised learning aims to reduce classification error.Phase 4—Selection of the Optimal CT: Selecting the optimal CT requires an internal validation system. This can be achieved by randomly splitting the sample into a training set and a validation set, or by applying cross-validation techniques. Cross-validation divides the dataset into subsets—e.g., 10 partitions, using 9 for training and 1 for validation in a recursive process.

A final CT includes the classification rules that generate a probability for the event of interest, such as mortality or disease diagnosis.

### 1.3. Use of Classification Trees in Medicine

CTs have been used in medicine since their inception. The main tasks assigned to them include generating classification rules for diagnosis, selecting variables based on their importance, determining cut-off points for continuous variables, and identifying clinical relationships among variables [6]. CT algorithms select the most relevant variables, their order of appearance in tree branching, and the optimal cut-off points [3]. The interpretability of classification rules makes CTs attractive for use in clinical settings [7].

A review of bibliographic databases reveals an exponential increase in publications using CT methodology, supporting their ongoing relevance in medical problems [7]. In the past two decades, the widespread adoption of machine learning techniques, including CTs, has further promoted their use [8].

We reference several studies that develop risk models using CTs and provide clear explanations of their methodological construction, such as a model for serious fall injury in older adults [9], or risk stratification in critically ill patients [10]. CTs have also been applied in nutrition, such as in malnutrition detection [11], identifying the relationship between frailty and diet quality indicators [12], or predicting dropout from psychological treatment in bariatric surgery candidates [13].

### 1.4. Types of Classification Trees

There are many types of CTs [2,3]. Broadly, they can be divided into three main types (see Table 1): simple models that generate a single CT, ensemble models that use multiple CTs to improve accuracy, and hybrid models that combine CTs with other machine learning techniques, such as fuzzy logic or artificial neural networks [14,15].

Different types of simple CTs vary based on their stopping criteria, pruning methods, and procedures for selecting the optimal tree [10]. The most commonly used simple models include CART, CHAID, C4.5, and ctree [16,17,18,19]. Table 1 shows their specifications and the available software for implementation [20,21,22,23]. Notably, CART-type trees perform well with small datasets, which explains their continued use in limited data scenarios [10].

Other types of simple CTs, such as FACT (Fast and Accurate Classification Tree), QUEST (Quick Unbiased and Efficient Statistical Tree), CRUISE (Classification Rule with Unbiased Interaction Selection and Estimation), GUIDE (Generalized Unbiased Interaction Detection and Estimation), and Bayesian trees, have shown specific advantages in particular patient groups [3].

**Table 1 nutrients-17-01903-t001:** Types of classification trees.

1. Simple Classification Trees				
	**CART**	**CHAID**	**C4.5**	**ctree**
**Description**	Classification and regression tree	Chi-square automatic interaction detection	Concept Learning Systems Version 4.5	Conditional inference trees
**Developer**	Breiman (1984) [16]	Kass (1980) [17]	Quinlan (1993) [18]	Hothorm (2006) [19]
**Primary Use**	Many disciplines with few data	Applied statisticians	Data miners	Applied statisticians
**Splitting Method**	Entropy Gini index	Chi-square tests F test	Gain ratio	Asymptotic approximations
**Branch Limitations**	Best binary split	Number of values of the input	Best binary split	Bonferroni-adjusted *p*-values
**Pruning**	Cross-validation	Best binary split *p*-value	Misclassification rates	No pruning
**Software ***	AnswerTree WEKA R-Python	AnswerTree R-Python	WEKA R-Python	R-Python
**2. Ensembled Classification Trees**				
	**Random Forest**	**AdaBoost**	**XG-Boost**	
**Description**	Uncorrelated forest	Adaptive boosting	Extreme gradient boosting	
**Developer**	Breiman and Cutler (2001) [24]	Freund and Schapire (1995) [25]	Chen and Guestrin (2016) [26]	
**Ensembled Method**	Parallel bagging	Adaptive boosting	Boosting gradient descent	
**Software ***	R/Python/Java	R/Python/Java	R/Python/Java	
**3. Hybrid Models**				
	**Fuzzy Random Forest**	**Random Forest Neural Network**		
**Other method**	Fuzzy logic	Neural network		
**Developer**	Olaru (2003) [15]	Khozeimeh (2022) [14]		
**Software ***	C language	Python		

Modified from [2,3,5,7]. * References of available software [20,21,22,23].

In general, model fitting parameters—called hyperparameters—include the tree’s maximum depth (branching levels), the minimum number of cases per terminal node, and the internal validation method used (e.g., training/validation split or cross-validation). Each simple CT model may require specific additional hyperparameters for proper functioning [2,3,9].

Hybrid models that combine CTs with other machine learning techniques are also an expanding area. A separate, dedicated review would be necessary to explore their methodologies in depth. In Table 1, we included only two examples of hybrid models: one that combines fuzzy logic with CTs [15], and another that incorporates artificial neural networks [14].

### 1.5. Advantages and Disadvantages of Classification Trees

Although we have already mentioned several advantages and disadvantages of CTs, they can be summarized as follows [2,3,4,5,6]:

#### 1.5.1. Advantages

Non-parametric models.Can handle all variable types (continuous, ordinal, categorical).Easy to interpret, with clinically meaningful classification rules.No additional calculations required to determine individual patient risk.Perform variable selection and establish variable hierarchy.Identify optimal cut-off points for continuous variables.Detect relationships among variables without assuming independence.Less affected by outliers or missing values.

#### 1.5.2. Disadvantages

Risk of overfitting and limited generalizability.High sensitivity to data, leading to model instability.Complex trees may lose interpretability.Require specific software and development methodology.Many CT types exist, and the most suitable one for a specific problem may not be obvious in advance.

In summary, it is necessary to strike a balance between the advantage of interpretability through classification rules and the methodological rigor required for their use.

## 2. ENPIC Study and the Obesity Paradox

We next describe the dataset used in this study, conducting a post hoc analysis of the ENPIC (Evaluation of Practical Nutrition Practices in the Critical Care Patient) study, applying classification tree (CT) methodology to explore the obesity paradox [27].

### 2.1. The ENPIC Study

Patients admitted to intensive care units (ICUs) often require artificial nutrition. The goal of artificial nutrition is to achieve optimal caloric and protein intake. Studies such as ENPIC aim to address questions related to improving this process [28,29,30].

The objective of the ENPIC study was to evaluate compliance with recommendations for specialized nutritional–metabolic therapy in ICU patients and to assess the impact of nutritional therapy on mortality in this population [29]. This was a multicenter study involving a sample of 525 ICU patients who required artificial nutrition, either enteral or parenteral. 

One of this study’s conclusions was that we are far from achieving caloric and protein goals in these patients, particularly in those with obesity, which negatively influences outcomes, including increased mortality [28,29,30].

### 2.2. The Obesity Paradox

Obesity is generally considered a risk factor for numerous medical conditions. However, the idea of a paradoxical protective effect of obesity in certain diseases has been discussed for over 20 years. The concept of the “obesity paradox” was first described in patients undergoing percutaneous coronary interventions, where improved survival was observed in overweight and obese individuals [31]. The obesity paradox suggests that obesity may confer a protective effect in some clinical contexts, leading to better survival outcomes [31]. Even a meta-analysis in patients with heart failure found some evidence supporting this paradox [32].

Its presence in ICU patients remains controversial [33]. Some authors argue that the phenomenon may be attributable to selection bias and confounding, arising from inadequate adjustment for variables influencing the obesity–mortality relationship [34,35,36].

A study published in this journal using the ENPIC dataset—Nutrition Therapy in Critically Ill Patients with Obesity—reported differences among BMI groups in some of the analyzed characteristics, as shown in Table 2 [27].

This table presents the general characteristics, nutritional therapy, and outcomes of ICU patients stratified by body mass index (BMI) categories. It reveals demographic differences across BMI groups (normal weight, overweight, and obese). There is a non-significant trend toward lower 28-day mortality in obese patients, raising the question of whether the paradox might be present in this cohort.

The findings suggest that building a mortality risk model requires adjustment for patient-related factors (age, sex, BMI group), nutritional status based on subjective global assessment (SGA), disease severity according to the APACHE II (Acute Physiology and Chronic Health disease Classification System II) score [37], and daily caloric (Kcal/kg/day) and protein (g/kg/day) intake. If the paradox is present, obesity would emerge as a protective factor.

We performed a multiple logistic regression (LR) model using the selected variables and calculated odds ratios (95% CI). Table 3 shows the model, which indicates that factors independently associated with 28-day mortality are older age, higher APACHE II score, malnutrition, and lower protein intake.

The obesity paradox was not detected in this model, as the odds ratio for obesity was not statistically significant. Given the limited sample size and the LR model used, we cannot confirm the presence of the paradox. Classification trees may uncover interactions or relationships not easily detected through traditional regression methods.

In the following sections, we apply CTs to the ENPIC dataset to illustrate methodological aspects and explore the potential presence of the obesity paradox.

## 3. Use of Classification Trees to Determine Cut-Off Points

Continuous variables cannot be arbitrarily categorized. Typically, literature-based criteria or statistical methods are used to justify selected cut-off points.

Classification trees can be used to identify cut-off points in continuous variables and convert them into categorical ones—for example, to define score thresholds for diagnosing heart failure in patients with pleural effusion [38].

Other examples in nutrition-related contexts include determining a TNF-α (Tumor Necrosis Factor-alpha) cut-off related to HbA1c levels using a CHAID tree [39] or defining walking speed thresholds indicative of severe mobility limitations in sarcopenic patients using a CART tree [40].

In our paradox example, we employed a CHAID tree to determine cut-off points for age and APACHE II score. Figure 1 shows that the selected age and APACHE II groups for categorization can be justified using classification trees.

## 4. Use of Classification Trees to Identify Relationships Between Variables

As mentioned earlier, one advantage of classification trees is their ability to detect relationships between variables. These relationships are often non-linear. Identifying such associations without assuming independence—and grouping patients based on these patterns—is essential for understanding how risk factors operate differently across subgroups [33].

In our example, previous research has suggested that exploring the obesity paradox requires stratifying patients by nutritional status [33]. In Figure 2, using a CART model, we observe that the risk of mortality is influenced by malnutrition. Among patients without malnutrition, the obese subgroup shows lower mortality, which may suggest that the paradoxical effect is detectable only in non-malnourished patients.

## 5. Multivariable Risk Models Using Classification Trees

A multivariable classification tree (CT) model can select only the most informative variables, establish a hierarchy, determine optimal cut-off points to categorize continuous variables, and indicate relationships between variables, resulting in a set of classification rules [7,10].

In the obesity paradox example, we included in the CT the same variables used in the logistic regression (LR) model (see Table 2). The CART-type classification tree model discarded caloric intake and sex. Figure 3 shows the classification tree with its classification rules. The first variable selected by the algorithm is age; the subsequent branches include malnutrition status, disease severity, protein intake, and BMI groups. Although not directly related to the paradox, it is interesting to note that in the subgroup of older patients with malnutrition, higher protein intake is associated with lower mortality.

Also highlighted in a red circle is a subgroup of patients in which a paradoxical effect appears to be present. An analysis of this specific group of 62 patients—23 obese and 39 non-obese—found statistically significant differences only in mortality (21.7% vs. 48.7%, *p* = 0.035) and the prevalence of hypertension (78.3% vs. 51.3%, *p* = 0.035). No significant differences were observed in age, sex, patient type, time to initiation of artificial nutrition, severity level, type of nutritional support (enteral or parenteral), or caloric and protein intake. This subgroup shows a significant reduction in mortality among obese patients compared to non-obese, suggesting the obesity paradox may exist in specific contexts. The conclusion of this CT model is that the obesity paradox does not exist in general but may be observed in specific patient groups, which can be identified using CT-based methodologies.

## 6. Ensemble Classification Tree Models

As previously noted, one disadvantage of simple CT models is their tendency to overfit, which limits their generalizability. To mitigate this issue, the idea arose of using not just one CT but rather an ensemble of trees to improve both the precision and generalizability of risk models [3].

Ensembles of CTs can be built in various ways (see Table 1), with the two most common being bagging and boosting [3,41].

Bagging (bootstrap aggregating) uses parallel ensemble learning to reduce model variance, averaging the results of individual CTs to produce a final prediction [42]. Each individual tree operates independently, allowing for fast performance. A representative of this method is the random forest algorithm [24,43].

In contrast, boosting involves sequential ensemble learning, where each CT is dependent on the previous ones. Boosting reduces bias and is considered “slow learning” compared to bagging. By adding one CT after another, the algorithm progressively improves classification. Popular boosting-based ensemble models include AdaBoost and gradient-based methods like XGBoost [44,45]. Boosting’s flexibility and strong performance have made it one of the most widely used techniques in recent years [45].

Although ensemble models outperform single-tree models in prediction accuracy, they come at the cost of reduced interpretability. This “black-box” nature has led to the development of methods to explain how these models work, providing clinicians with insights into variable importance and relationships [45].

One of the most used explanatory techniques is SHAP (SHapley Additive exPlanation) values. Based on cooperative game theory, SHAP values determine the contribution of each “player” (variable) to the final model outcome. A SHAP value reflects each variable’s contribution to the prediction, and results are visualized using importance plots. SHAP also generates partial dependence plots showing the relationship between each variable and the model outcome [46].

In the obesity paradox example, we use an XGBoost and SHAP model for explanation. Using ensemble methods like XGBoost involves tuning many hyperparameters, which increases complexity [47]. Some of the hyperparameters used in our XGBoost model included learning rate (eta) set at 0.3, number of trees (n_estimators) set at 20, gamma (loss reduction threshold) at 0, and maximum tree depth (max_depth) at 6.

The same variables used in the LR and CART models were included. Figure 4 shows the SHAP values.

The graph provides a visual explanation of the model, showing the influence of each variable on the studied outcome, which in this case is mortality. It displays the relative importance of variables based on their SHAP scores. The association with mortality is also indicated, with values on the right linked to higher mortality and those on the left to lower mortality. The variable categories are color-coded, enabling the identification of clear groupings associated with increased or decreased risk of the event.

The XGBoost model identified age as the most important variable, followed by caloric intake (higher intake = worse outcome), protein intake (higher intake = better outcome), and BMI group. Within the BMI analysis, the obese group showed an association with improved survival.

Figure 5 shows the partial dependence plots. These graphs illustrate the individual impact of each factor on the studied outcome. For example, the age groups demonstrate that higher age corresponds to higher SHAP values, indicating a stronger association with mortality. In contrast, the BMI groups reveal that obese individuals (highlighted with a red circle) have lower SHAP values, indicating a weaker association with mortality.

Both figures indicate that being obese is associated with a protective effect on mortality.

## 7. Model Evaluation

The methodology for evaluating a risk model is also very extensive and could, by itself, constitute an independent review. However, it is important to note that the development, validation, and assessment of the predictive accuracy of a risk model should follow standardized procedures, such as the TRIPOD (Transparent Reporting of a Multivariable Prediction Model for Individual Prognosis or Diagnosis) guidelines, which provide guidance on all the necessary steps from model construction to evaluation [48].

Our LR models, the CART-type decision tree, and the XGBoost ensemble model generate a probability of death for each patient. We can assess model accuracy using global performance metrics (Brier score, accuracy, recall, f-measure), discrimination (AUC ROC), and calibration (calibration curve with slope and intercept values) [10,43,49].

For example, Figure 6 illustrates the discriminatory ability of the models through the ROC curve and the corresponding area under the curve (AUC). We observe that the XGBoost ensemble model achieves the highest AUC, while the LR and CART models yield similar values.

## 8. Discussion

In this article, we describe how classification trees (CT) can be used in clinical research as a complementary tool to traditional statistical techniques.

Using the obesity paradox as an example, we explored the methodology based on classification trees. One of the main advantages of simple CTs is their interpretability [10]. This feature, combined with the ability to identify patient subgroups—defined by classification rules—with distinct characteristics, makes them appealing to clinical practitioners [2,3].

However, a key disadvantage of CTs is the risk of overfitting and their strong dependence on the specific dataset used for model construction, which can lead to instability issues [6]. To address these limitations, ensemble CTs have been developed, achieving higher accuracy and better generalization performance [24,25,26]. Nevertheless, this improved performance often comes at the cost of reduced interpretability—an issue also common in more complex machine learning methods and other artificial intelligence tools [49,50].

Although complex models tend to lack transparency, several techniques have emerged to help interpret their inner workings. In our work, we applied SHAP (SHapley Additive exPlanations), which provides insights into variable importance and reveals partial relationships between individual risk factors and the predicted outcome [51,52].

Our work is not without limitations. The depth of the methodological review could have been greater, although we believe that the references provided may help interested readers to achieve that objective. The example used to illustrate the methodological review involved a simple selection of variables to facilitate the generation, interpretation, and explanation of the classification tree (CT) models. In model construction, the inclusion of more variables or the use of alternative methodologies—both simple and ensemble-based—could have led to different results and conclusions. Moreover, the models we have developed should undergo external validation to demonstrate their generalizability.

This suggests that CTs are especially useful in uncovering clinically relevant patterns that could guide personalized medicine approaches. Nevertheless, caution should be taken when interpreting these findings, as tree-based models are sensitive to data characteristics and may require validation in independent cohorts to confirm their generalizability.

We can state that classification tree–based methodologies remain an important area of experimentation and are constantly evolving technologically [53,54]. Nutrition professionals working with risk models should consider the use of CTs. Simple trees remain appealing due to their interpretability, while more sophisticated models must continue to evolve to become more user-friendly [55]. We also offer our collaboration, sharing our experience with other research groups interested in developing CT-based models.

## 9. Conclusions

The application of CT methodology in our example, based on the ENPIC study database, has allowed for us to illustrate several important aspects:How CTs can be used to establish cut-off points for continuous variables.How they can identify interactions between variables that might go unnoticed in traditional regression models.How multivariable CT models generate decision rules and stratify patient risk based on the most influential predictors.How ensemble methods such as random forest and XGBoost improve predictive accuracy.And finally, how explanatory tools like SHAP values can provide insight into the structure and predictions of complex models.

Although we did not find strong evidence to confirm the existence of the obesity paradox in critically ill patients in a general sense, CT-based analysis helped us identify a specific subgroup of patients in which obesity appeared to be associated with reduced mortality.

In conclusion, CTs—especially when combined with ensemble methods and model explanation techniques—are valuable tools in clinical epidemiology. They can support both predictive modelling and the exploration of complex relationships among clinical variables, offering new perspectives to improve patient care.

## Figures and Tables

**Figure 1 nutrients-17-01903-f001:**
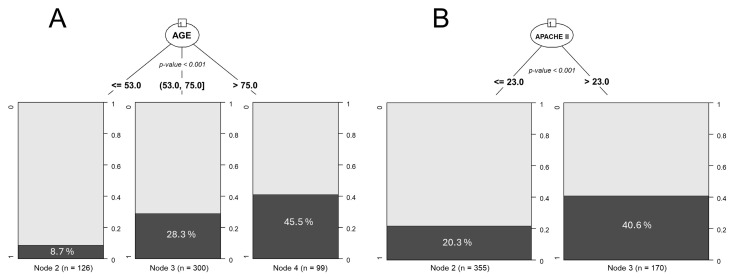
Using CHAID-type AC trees to establish cut-off points for continuous variables. (**A**) Cut-off points for age. (**B**) Cut-off points for the APACHE II severity score variable. R software (CHAID library) was used [23].

**Figure 2 nutrients-17-01903-f002:**
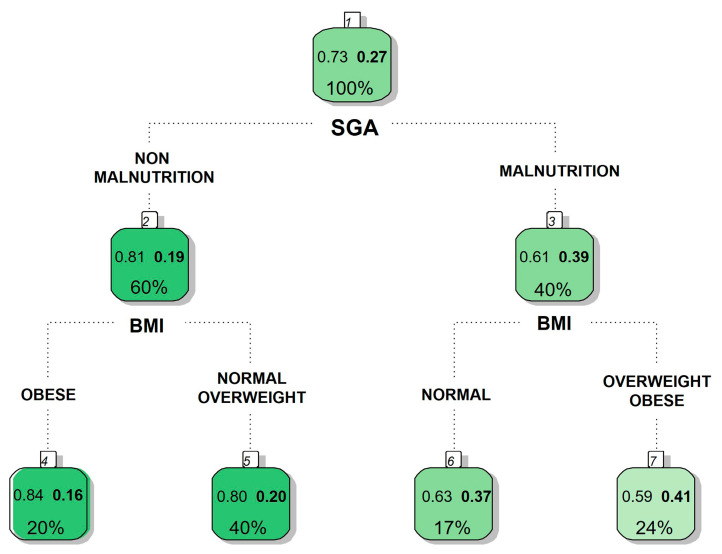
CART-type AC model to establish the relationship between two variables. Values in bold indicate 28-day mortality. SGA: subjective global assessment; BMI: body mass index group. It is observed that the non-malnutrition group of obese patients has a lower mortality rate. R software (rpart library) was used [23].

**Figure 3 nutrients-17-01903-f003:**
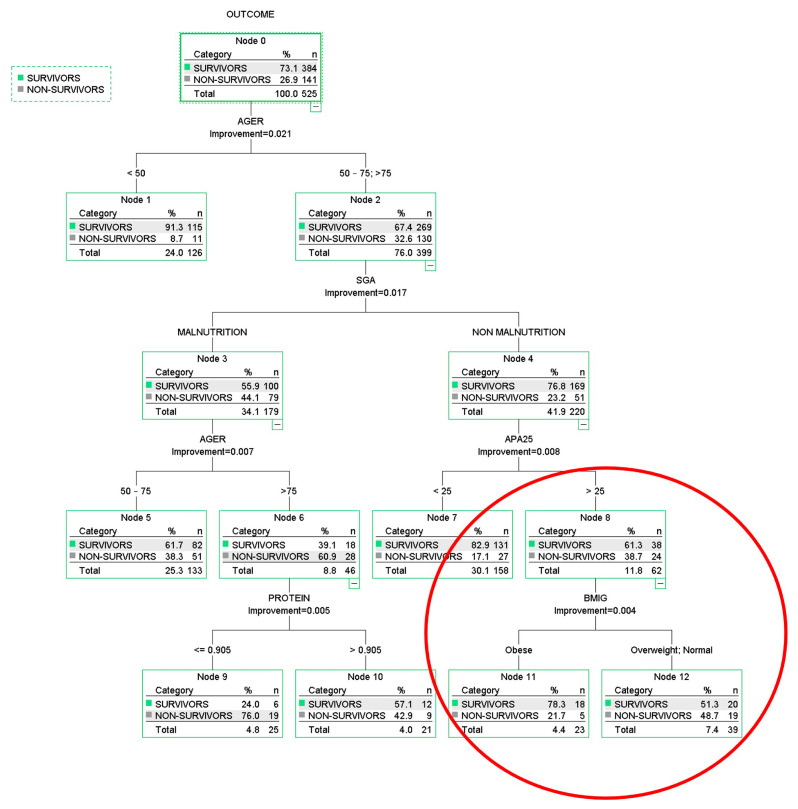
Multivariate model based on CART type. AGER: age groups; SGA: subjective global assessment; APA25: APACHE II score greater than 25; BMIG: body mass index group; PROTEIN: protein intake (g/kg/day). The hierarchy of variables is shown, and a special group in which obese patients have a lower mortality rate than those with overweight or a normal BMI is indicated in the red circle. AnswerTree 3.0 software was used [24].

**Figure 4 nutrients-17-01903-f004:**
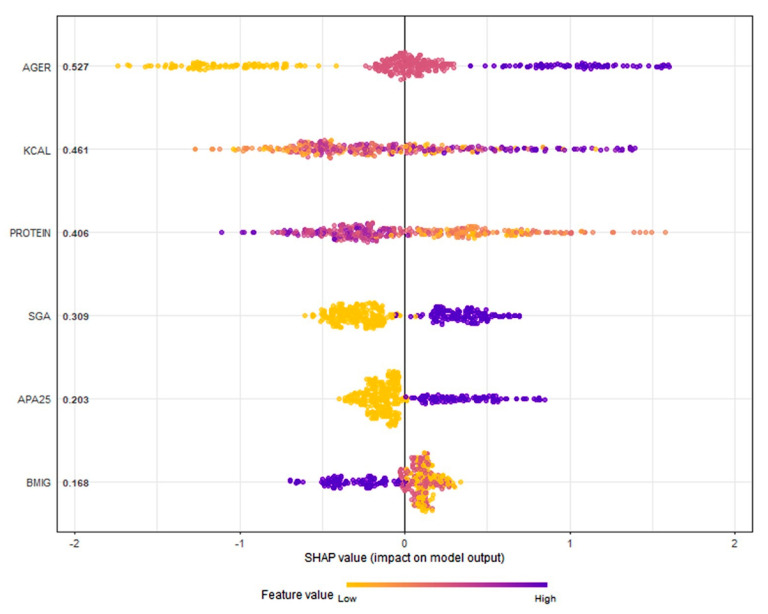
Graph showing SHAP scores generated by the XGBoost model. AGER: age groups; KCAL: calorie intake (Kcal/kg/day); PROTEIN: protein intake (g/kg/day); SGA: subjective global assessment; APA25: APACHE II score greater than 25; BMIG: body mass index group. The importance of each variable can be observed, with age being the greatest. It is also observed that the obese patient group has a negative impact on mortality, suggesting a paradoxical effect of obesity. R software (xgboost and SHAPforxgboost libraries) was used [23].

**Figure 5 nutrients-17-01903-f005:**
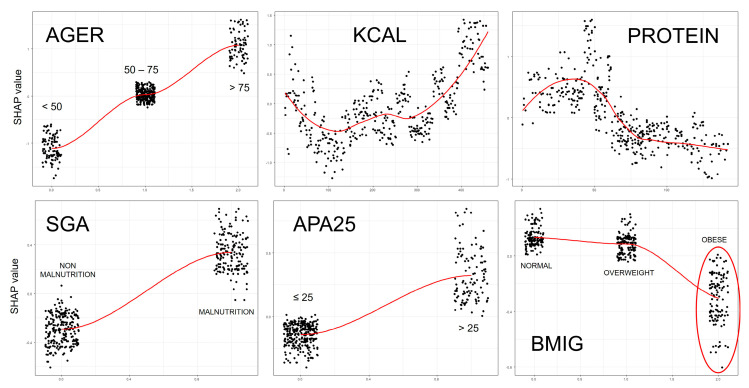
Partial dependency graphs between each variable and mortality. AGER: age groups; KCAL: calorie intake (kcal/kg/day); PROTEIN: protein intake (g/kg/day); SGA: subjective global assessment; APA25: APACHE II score greater than 25; BMIG: body mass index groups. We observed different behavior between calorie and protein intake. Within the BMI groups, mortality was lower in the obese group. R software (SHAPforxgboost library) was used [23].

**Figure 6 nutrients-17-01903-f006:**
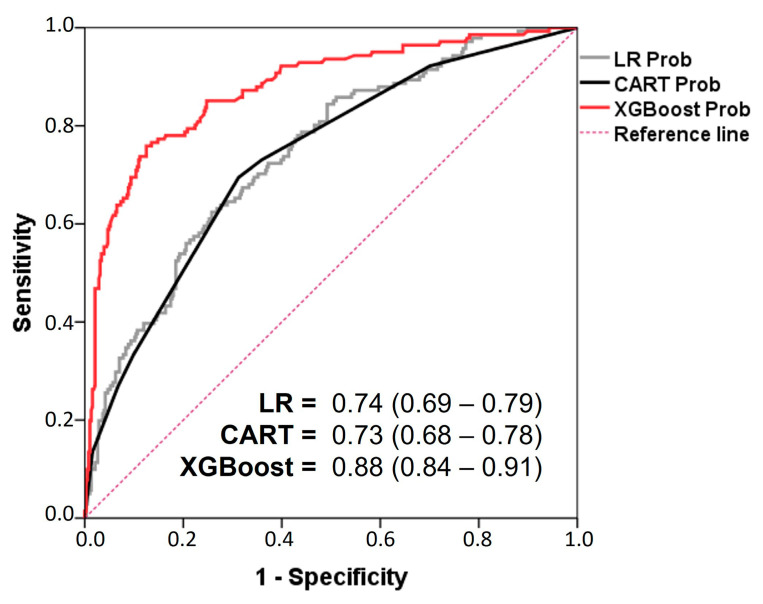
ROC analysis of the developed models. The area under the ROC curve values are shown. LR: logistic regression model; CART: CART-type classification tree; XGBoost: XGBoost-type ensemble classification tree model; Prob: probability of death at 28 days. The LR and CART-type AC models had similar AUCs. The XGBoost model achieved better discrimination (Long’s test with *p* < 0.001).

**Table 2 nutrients-17-01903-t002:** General characteristics, nutritional therapy, and outcomes of patients admitted to the ICU based on body mass index subgroup. Results of the ENPIC study [20].

		All Patients *n* = 525	Normal *n* = 165	Overweight *n* = 210	Obese *n* = 150	*p*-Value
**Baseline characteristics and comorbidities**						
Age, years, mean ± SD		61.5 ± 15	58.8 ± 16.5	62.8 ± 14.7	62.7 ± 13.5	0.05
Sex, male patients, *n* (%)		67.2% (353)	64.8% (107)	74.8% (157)	59.3% (89)	**0.003 ^B^**
Hypertension, *n* (%)		43.6% (229)	33.9% (56)	41.9% (88)	56.7% (85)	**0.01 ^A,B^**
Diabetes mellitus, *n* (%)		25% (131)	21.2% (35)	20% (42)	36% (54)	**0.001 ^A,B^**
AMI, *n* (%)		14.1% (74)	8.5% (14)	16.7% (35)	16.7% (25)	**0.04 ^B^**
Neoplasia, *n* (%)		20.6% (108)	24.2% (40)	19.5% (41)	18% (27)	0.11
Type of patient	Medical, *n* (%)	63.8% (335)	65.5% (108)	62.9% (132)	63.3% (95)	0.81
	Trauma, *n* (%)	12.6% (66)	10.9% (18)	15.2% (32)	10.7% (16)	0.75
	Surgery, *n* (%)	23.6% (124)	23.6% (39)	21.9% (46)	26% (39)	0.67
**Prognosis ICU scores and nutrition status on ICU admission**						
APACHE II, mean ± SD		20.3 ± 7.9	19.7 ± 7.6	20.1 ± 7.5	21.2 ± 8.5	0.18
Malnutrition (based on SGA), *n* (%)		41% (215)	52.7% (87)	37.1% (78)	33.3% (50)	**0.01 ^B^**
**Characteristics of Medical Nutrition Therapy**						
Early nutrition, <48 h, *n* (%)		74.9% (393)	77.6% (128)	75.2% (158)	71.3% (107)	0.43
Kcal/kg/day, mean ± SD		19 ± 5.6	23.1 ± 6	18.6 ± 3.7	15.27 ± 4.24	**0.001 ^A^**
Protein, g/kg/day, mean ± SD		1 ± 0.4	1.2 ± 0.4	1 ± 0.3	0.8 ± 0.2	**0.01 ^A,B^**
EN		63.2% (332)	59.4% (98)	64.3% (135)	66% (99)	0.34
PN		15.4% (81)	13.3% (22)	16.2% (34)	16.7% (25)	0.85
EN-PN		7.8% (41)	8.5% (14)	7.6% (16)	7.3% (11)	0.92
PN-EN		13.5% (71)	18.8% (31)	11.9% (25)	10% (15)	0.27
**Outcomes**						
Mechanical ventilation, *n* (%)		92.8% (487)	89.1% (147)	93.8% (197)	95.3% (143)	0.08
Vasoactive drug support, *n* (%)		77% (404)	73.9% (122)	79.5% (167)	76.7% (115)	0.44
Renal replacement therapy, *n* (%)		16.6% (87)	16.4% (27)	12.9% (27)	22% (33)	0.07
ICU stay, days, mean ± SD		20.3 ± 18	18.2 ± 13.8	21.1 ± 17.1	21.6 ± 22.5	0.08
28-day mortality, *n* (%)		26.7% (140)	29.1% (48)	27.1% (57)	23.3% (35)	0.51

AMI: acute myocardial infarction; PN: parenteral nutrition; EN: enteral nutrition; SD: standard deviation; APACHE II: Acute Physiology and Chronic Health Disease Classification System II; SGA: subjective global assessment; ICU: intensive care unit. Statistically significant *p*-values are written in bold. Statistical results correspond to ANOVA *p* values. Bonferroni post hoc testing with statistically significant differences ^A^ between normal weight and obese subgroup; ^B^ between overweight and obese subgroup.

**Table 3 nutrients-17-01903-t003:** Multiple logistic regression model (LR) with 28-day mortality as the outcome variable.

Variable	OR (95% CI)	*p*-Value
Age (years)		
<50	1	
50–75	3.3 (1.7–6.5)	0.001
>75	7.0 (3.3–14.9)	<0.001
Sex (Male)	1.0 (0.6–1.6)	0.998
APACHE II score		
≤25	1	
>25	2.2 (1.4–3.5)	<0.001
BMI groups		
Normal	1	
Overweight	0.9 (0.5–1.5)	0.623
Obese	0.7 (0.4–1.4)	0.651
SGA		
Non-malnutrition	1	
Malnutrition	2.6 (1.74.0)	<0.001
Median Kcal/Kg/day	1.1 (0.9–1.1)	0.315
Median g protein/Kg/day	0.3 (0.1–0.9)	0.022

OR: odds ratio; CI: confidence interval; APACHE II: Acute Physiology and Chronic Health Disease Classification System II; BMI: body mass index; SGA: subjective global assessment.

## Data Availability

The data in this study are available on request from the author.

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
