# Peer review of "Methodological Review of Classification Trees for Risk Stratification: An Application Example in the Obesity Paradox"

_nutrients, 2025, doi:10.3390/nu17111903_

Round 1
Reviewer 1 Report
Comments and Suggestions for Authors
This is an interesting review article concerning the classification trees for risk stratification and its implication in obesity paradox. This topic has adequate novelty; however, some points should be addressed.
- Please explain in the abstract the abbreviations: ICU, CART, CHAID, XGBoost.
- The authors should add some statements at the beggining of the introduction section before reporting the aim of their manuscript.
- In section 1.1, concerning the statements "The earliest references to CTs are attributed to Quinlan in 1986 [3]. Regression trees, which use continuous outcome var-iables, had already been in use for over 50 years by then [2].", the authors should briefly add some characteristic examples.
- Table 1 should be transferred after its citation into the text.
- In Table 1, the number of references of developers, e.g. Breiman et al., 1984 [14] should be included.
- In section 2.1, please specidy the abbreviation for ICU patients.
- In section 2.1, the authors report the phrasing :critically ill patients". This should be further specified.
- In section 2.2, the obesity paradox should be more deeply described.
- Table 3 should be cited into the text before the table.
- The resolution of all figures should be improved.
- In section 3, please report the full abbreviation of TNF-α
- Discussion section is too short. The authors should try to enrich the discussion section.
Author Response
Reviewer-1
This is an interesting review article concerning the classification trees for risk stratification and its implication in obesity paradox. This topic has adequate novelty; however, some points should be addressed.
- Please explain in the abstract the abbreviations: ICU, CART, CHAID, XGBoost.
- The clarification regarding the abbreviations has been included.
- The authors should add some statements at the beginning of the introduction section before reporting the aim of their manuscript.
- An introductory paragraph has been added, including a reference to decision-making in medicine.
- In section 1.1, concerning the statements "The earliest references to CTs are attributed to Quinlan in 1986 [3]. Regression trees, which use continuous outcome variables, had already been in use for over 50 years by then [2].", the authors should briefly add some characteristic examples.
- Examples on general medical topics and specific issues related to nutrition have been added in Section 1.3.
- Table 1 should be transferred after its citation into the text.
- Table 1 has been relocated to appear after its first citation in the text.
- In Table 1, the number of references of developers, e.g. Breiman et al., 1984 [14] should be included.
- All bibliographic references have been added within Table 1.
- In section 2.1, please specify the abbreviation for ICU patients.
- The abbreviation has been specified.
- In section 2.1, the authors report the phrasing :critically ill patients". This should be further specified.
- The objective of the study and the population included have been clarified.
- In section 2.2, the obesity paradox should be more deeply described.
- The concept of the obesity paradox has been further elaborated.
- Table 3 should be cited into the text before the table.
- The table has been repositioned.
- The resolution of all figures should be improved.
- The figures lost resolution when integrated into the Word format. The versions included in the final article format will have higher resolution.
- In section 3, please report the full abbreviation of TNF-α
- The abbreviation has been defined and included in the abbreviations section of the article.
- Discussion section is too short. The authors should try to enrich the discussion section.
- The discussion section has been revised. It has been expanded and now includes additional references explaining complex models. The limitation regarding the lack of external validation of the developed models has also been added.
Reviewer 2 Report
Comments and Suggestions for Authors
This manuscript presents a well-structured and pedagogically oriented overview of classification trees (CTs), accompanied by an example application using data from the ENPIC study to explore the so-called “obesity paradox.” This work is timely and methodologically informative, particularly for clinical researchers seeking to understand or adopt CT-based approaches. However, some areas require clarification and refinement to meet the standards of a methodological review article in Nutrients.
Main comments
The review aims to be accessible to clinicians, which is commendable. However, the degree of technical depth is uneven. While the section on CART, CHAID, and ensemble methods is solid, the discussion of hybrid models (e.g., fuzzy logic + neural networks) is superficial and lacks concrete examples or references to biomedical applications.
The ENPIC study is a relevant example, but the analysis does not fully exploit the strengths of tree-based models. Discussion on:
- Generalizability of the model.
- Overfitting problems.
- Comparative performance between subgroups (e.g., by sex, comorbidities).
Although internal validation (e.g., cross-validation) is addressed, external validation is not.
Minor comments
Terms such as “classification tree,” “CART,” and “decision tree” are used interchangeably. Clarify distinctions where relevant, particularly for nonstatistical readers.
When interpreting SHAP plots and partial dependence plots, stronger guidance on how to interpret directionality and strength of association would benefit the clinical reader.
Figures (e.g., tree diagrams, SHAP plots) are informative, but could benefit from clearer legends and explanation of axes, especially in the case of partial dependence plots.
Although the references are extensive and generally appropriate, some citations are outdated. For example, recent literature on XAI (explainable AI) in clinical prediction could strengthen the interpretive section.
Author Response
Reviewer-2
This manuscript presents a well-structured and pedagogically oriented overview of classification trees (CTs), accompanied by an example application using data from the ENPIC study to explore the so-called “obesity paradox.” This work is timely and methodologically informative, particularly for clinical researchers seeking to understand or adopt CT-based approaches. However, some areas require clarification and refinement to meet the standards of a methodological review article in Nutrients.
Main comments
The review aims to be accessible to clinicians, which is commendable. However, the degree of technical depth is uneven. While the section on CART, CHAID, and ensemble methods is solid, the discussion of hybrid models (e.g., fuzzy logic + neural networks) is superficial and lacks concrete examples or references to biomedical applications.
We have added examples of hybrid models. A deeper analysis of these models would require a separate review.
The ENPIC study is a relevant example, but the analysis does not fully exploit the strengths of tree-based models. Discussion on:
- Generalizability of the model.
- Overfitting problems.
- Comparative performance between subgroups (e.g., by sex, comorbidities).
Although internal validation (e.g., cross-validation) is addressed, external validation is not.
The discussion section has been revised. It has been expanded and now includes additional references explaining complex models. The limitation regarding the lack of external validation of the developed models has also been added.
Minor comments
- Terms such as “classification tree,” “CART,” and “decision tree” are used interchangeably. Clarify distinctions where relevant, particularly for nonstatistical readers.
- The term classification has been used consistently throughout the text as a reference.
- When interpreting SHAP plots and partial dependence plots, stronger guidance on how to interpret directionality and strength of association would benefit the clinical reader.
- The interpretation of Figures 5 and 6 has been clarified.
- Figures (e.g., tree diagrams, SHAP plots) are informative, but could benefit from clearer legends and explanation of axes, especially in the case of partial dependence plots.
- The interpretation of Figures 5 and 6 has been clarified.
- Although the references are extensive and generally appropriate, some citations are outdated. For example, recent literature on XAI (explainable AI) in clinical prediction could strengthen the interpretive section.
- Additional references have been included in the discussion section.
Round 2
Reviewer 1 Report
Comments and Suggestions for Authors
The authors have significantly improved their manuscript.